# Investigation of the Hue–Wavelength Response of a CMOS RGB-Based Image Sensor

**DOI:** 10.3390/s22239497

**Published:** 2022-12-05

**Authors:** Hyeon-Woo Park, Ji-Won Choi, Ji-Young Choi, Kyung-Kwang Joo, Na-Ri Kim

**Affiliations:** 1Center for Precision Neutrino Research, Department of Physics, Chonnam National University, Yongbong-ro 77, Puk-gu, Gwangju 61186, Republic of Korea; 2Department of Fire Safety, Seoyeong University, Seogang-ro 1, Puk-gu, Gwangju 61268, Republic of Korea

**Keywords:** color space, H-W curve, digital camera, CMOS sensor

## Abstract

In this study, a non-linear hue–wavelength (H-W) curve was investigated from 400 to 650 nm. To date, no study has reported on H-W relationship measurements, especially down to the 400 nm region. A digital camera mounted with complementary metal oxide semiconductor (CMOS) image sensors was used. The obtained digital images of the sample were based on an RGB-based imaging analysis rather than multispectral imaging or hyperspectral imaging. In this study, we focused on the raw image to reconstruct the H-W curve. In addition, several factors affecting the digital image, such as exposure time or international organization for standardization (ISO), were investigated. In addition, cross check of the H-W response using laser was performed. We expect that our method will be useful as an auxiliary method in the future for obtaining the fluor emission wavelength information.

## 1. Introduction

In this study, we attempted to obtain a hue–wavelength (H-W) relationship by using a digital image. Color image processing was performed using a digital camera based on CMOS sensors covered by a Bayer color filter array (CFA). The photographed digital images were stored in trichromatic values. Red (R), green (G), and blue (B) filters were implemented on the CMOS sensor devices, and the data were saved in the form of raw RGB. The RGB space was defined by the Commission Internationale de l’Eclairage (CIE, International Commission on Illumination) in 1931 [1,2]. CIE RGB is the most common RGB space, and it is used today as a standard to define colors and as a reference for other color spaces such as a standard RGB (sRGB) or adobe RGB format through further processes. In addition, CIE also defined the XYZ color space, which was one of the first mathematically defined color spaces. By combing the (R, G, B) values, a broad array of colors in the visible region can be reproduced. Alternatively, hue (H), saturation (S), and value (V) are other representations of a color space. Once the RGB values are determined, they can be converted to HSV values. According to the HSV model, color is defined by a non-linear mathematical transformation rather than a simple combination of adding or subtracting the primary (R, G, B) colors [3]. Physically, H is directly related to wavelength. If the RGB values of each pixel in the digital image are known, then the H value information can be obtained, and the corresponding wavelength can be extracted. In this way, data in raw RGB format for each camera is saved, and data can also be saved in sRGB or adobe RGB format through pre-processing.

H is determined by the dominant wavelength of the (R, G, B) colors. The formula for converting RGB to HSV is as follows [2,3,4,5]:(1)$$H=\{\begin{array}{c} \theta \\ 360-\theta \end{array}\begin{array}{c} \mathrm{if}\;B\leq G\\ \mathrm{if}\;B>G \end{array}$$
(2)$$\theta =\cos^{-1}\left\{\frac{\frac{1}{2}\left[\left(R-G\right)+\left(R-B\right)\right]}{{\left[{\left(R-G\right)}^{2}+\left(R-B\right)\left(G-B\right)\right]}^{1/2}}\right\}$$
(3)$$S=1-\frac{3}{\left(R+G+B\right)}\left[\min\left(R,G,B\right)\right]$$
(4)$$V=\frac{1}{3}\left(R+G+B\right)$$

Our aim was to investigate the H-W relationship using the CMOS RGB-based image sensor technology. In previous studies conducted using CMOS or Foveon sensors [6,7,8], the H-W relationship was measured in the wavelength region from 500 to 650 nm. However, to date, no measurement has been reported down to the 400 nm region [9,10,11]. We are interested in this wavelength region because the optimal operating range of a photomultiplier tube (PMT) used in high-energy physics usually lies in this wavelength band. Therefore, we focused on the H-W curve itself to cover the whole range of wavelengths from 400 to 650 nm.

A digital image in color space was captured by a CMOS Bayer CFA-based digital camera. Each pixel usually only has three RGB bands in the visible spectrum. The three bands are limited to observing the scene in the photograph and generally do not provide sufficient information about the external or internal components of the material. To overcome these shortcomings, multispectral imaging (MSI) or hyperspectral imaging (HSI) were developed [12,13,14]. MSI collects data in a small number of spectral bands. HSI is a superset of MSI and is a continuous spectrum with a higher spectral resolution. Absorption, reflection, and fluorescence spectral data have been added to the HSI, allowing for a more accurate analysis and understanding of micro- and nanoscale spectral features. However, in this study, RGB analysis was enough to reconstruct the digital image.

## 2. Digital Image Sensor Technology

### 2.1. CMOS Bayer CFA and Foveon Image Sensor Technology

In our study, we used a digital single lens reflex (DSLR, Canon EOS D series, Tokyo, Japan) camera equipped with a CMOS image sensor. Each pixel of most commercial CMOS image sensors is covered by a CFA and receives only a specific range of wavelengths. In the CFA configuration, each pixel captures just one color among R, G, or B. The other two missing color values are estimated from the recorded mosaic data of the neighboring pixels. This interpolation process is called demosaicing (or demosaicking) [15,16,17]. Numerous demosaicing algorithms have been proposed, and one example is shown in Figure 1a. Furthermore, various integration technologies of CFA-based CMOS image sensor have been developed [18,19,20,21,22]. Nevertheless, due to the lack of information during the demosaicing process, the original color cannot be fully expressed [23].

As another configuration, Foveon X3 technology has been developed, as shown in Figure 1b. In the Foveon method, three photodetectors are stacked at each pixel location. Each of the photodetectors is tuned to capture a different color: one dedicated to red light, one to green, and one to blue. As a result, Foveon image sensors capture three colors at every pixel. Compared to the Bayer CFA, the Foveon image sensor delivers a very smooth color reproduction [7,8]. However, we did not use this image sensor technology in this study. As already mentioned, in the neutrino experiment, there is no need to distinguish the wavelength of light down to the few-nanometers scale. The optimal wavelength of a bi-alkali PMT is approximately 400~450 nm, and when the neighboring light enters, the PMT can generate a signal, which is subsequently amplified. Therefore, even if a specific wavelength is not used, the wavelength band value is sufficient for the image analysis.

Figure 1c shows the mapping between wavelength and hue for the Canon camera employing a CMOS sensor based on CFA technology [6,7,8]. This graph shows several features. There are plateau regions in the wavelength in range of 530~560 nm and over 600 nm. At the near end of wavelengths 530~560 nm, a step-like H-W response is evident. These features are a direct result of the CFA color filters used in the CMOS sensor. The CFA arrangement made these patterns in the H-W curve. At 120° hue (wavelength ~560 nm), only the pure green component exists. Because neither blue nor red filters transmit significantly in this region, the plateau naturally occurs. The sharp drop of hue to 0^o^ over wavelength 620 nm is also due to the properties of the CFA color filters. Since the nature of the H-W response of CFA color filters does not allow wavelengths of a few nanometer to be distinguished, it should be used with caution in certain wavelength ranges.

Figure 1d shows the H-W relationship of the Foveon image sensor technology [7,8]. The curve shows a smooth monotonic decrease in hue with wavelength. This provides a good relationship for wavelength discrimination, particularly between 500 and 600 nm. The thickness of the line includes uncertainties that appear when the amount of light entering all the cameras is adjusted. The H-W curve is not highly sensitive to the amount of light. The curve at the extreme end of the wavelength shows a peak due to the poor camera response at the infrared (IR) cut-off filter beyond 675 nm. Unlike the CMOS CFA technology, the Foveon curve did not show any step-like H-W response. This indicates that the H-W curve can be a good candidate to decode color images, especially those of a monochromatically illuminated particle using this image sensor.

### 2.2. Image Processing Pipeline and Raw Image Analysis

When taking a picture, the pipeline process plays a key role in creating a digital color image, as shown in Figure 2. An image pipeline is a set of components consisting of several distinct processing blocks. All the processes can be grossly classified into two categories, such as raw image and joint photographic experts group (jpeg) image. Initially, the raw image data are saved. Aperture, shutter speed, and international organization for standardization (ISO) number affect these raw data. Referring to a camera, ISO means standards of measurement. Raw images are the minimally processed data obtained from the CMOS image sensor of the camera. Basically, the raw data contain information about the amount of light captured by each pixel [24,25].

After the raw data are created, various processes are implemented to develop jpeg images. During the demosaicing process, a Bayer filter is applied to the data received by the camera. Since a pixel has only one value (R or G or B), all the three values are applied to all the pixels through interpolation. The missing color channels of the raw mosaiced images are estimated from a CFA to reproduce full-color images**.** This is an essential process for single-sensor digital cameras.

Since the RGB value obtained from the camera is the value from the sensor and not the real color seen by the human eye, it is necessary to process it to make it similar to the color seen by the human eye. The outline for this process is as follows:$$\;\;\;\;\;\;\;\;\;\;\;\;\;{\left[\begin{array}{c} \;R\;\\ \;G\;\\ \;B\; \end{array}\right]}_{\mathrm{sRGB}}=M{\left[\begin{array}{c} \;R\;\\ \;G\;\\ \;B\; \end{array}\right]}_{\mathrm{raw},\;}$$
where M represents a matrix for converting the raw RGB color space through the CIE XYZ color space to a standard RGB (sRGB) color space [26]. The above is expressed in matrix form to imply the conversion of raw RGB to sRGB. The sRGB is an international standard RGB color space adopted by all general software and hardware, such as displays, printers, and digital cameras. In this process, the color is reconstructed by adjusting the white balance; that is, the original raw data are reconstructed into a color suitable for a digital camera image output. The above conversion processes are performed through a (3 × 3) matrix. Simultaneously, the camera RGB values with negative values on the CIE XYZ color space are adjusted. In addition, brightness, contrast, saturation adjustment, gamma correction, and noise reduction are performed. Through these complicated processes, the RGB values are finally corrected to a color similar to that seen by the human eye. Finally, a jpeg image is created. The jpeg image represents a lossy compressed data format created by down-sampling, discrete Fourier integral transformation, and encoding processes. The R, G, and B values of each pixel are stored in a color look-up table with a scale of 256 in 8-bit digital camera case. However, the corrected color is only a similar color and is not exactly the same as the color visible to the human eye. For this reason, we used the raw data, which were sufficient for our analysis purposes.

## 3. Experimental Set-Up

### 3.1. Light Source and Diffraction Grating

Figure 3a shows a simple experimental setup for generating light and taking pictures with a digital camera. A light emitting diode (LED, manufacturer: JPL International CO. LIMITED, VL-110, Shenzhen, China) source that simultaneously generated wavelength in the whole range from 400 to 650 nm was used. An optical aperture and a collimator were used to maximize the light passing through the diffraction grating. Significant efforts were invested to prevent external background or stray lights from entering the camera lens. The position of the diffraction pattern changed depending on the wavelength of the incident light, as shown in Figure 3b. At both the ends, the intensity of light was weak, and thus, special care was needed. We measured a distance (Y) from the right central white image to the left each color band in Figure 3c. If we know this value, then we can obtain an angle (θ) between the normal line of the ray and the diffraction image. From this, we can calculate the wavelength using the formula dsinθ = mλ, where d is the diffraction grating slit separation, and m is the diffraction order with an integer. When m = 0, which is the center of the beam axis, there is no chromatic dispersion, and only the incident white light is visible. If the grating has N lines per meter, then the grating spacing is given by 1/N. Further, m is called the spectrum order. For m = 1, we have the first diffraction maximum. When drawing the H-W curve, the wavelength bin size was set to 2 nm. A V cut was applied to separate the diffraction pattern and the background. Finally, the H values were obtained from the diffraction images using the Equations (1)–(4).

### 3.2. Comparison of Theoretical Diffraction Curve and Experimental Measurement

Since it is crucial to determine the wavelength according to the distance, we checked it using laser modules with wavelengths of 440 and 638 nm. We compared the theoretical diffraction fringe positions with the actual laser module positions produced by the images. As shown in Figure 4, the angle can be precisely measured with almost no difference. Several factors can be considered as the cause of this small difference. First, it can originate from the inaccuracy of the diffraction grating spacing. Currently, a diffraction grating with 1 line per 1205 nm is used. Second, the measurement of the distance between the screen center and the diffraction grating may be inaccurate. In our case, this distance was maintained at 50 cm, and careful efforts were made to measure with the maximum possible precision.

## 4. Investigation on Various Factors Affecting H-W Curve

### 4.1. Captured Area and H Distribution

Figure 5 shows one of the H distribution in each region from the inside to the outside of the raw image. A1 region is the central part of the image, and it was bright compared to other region. As shown in Figure 5b, a tail on the left side was caused by the saturated pixels in the H distribution; thus, it was not used for the analysis. In Figure 5c,d, the H peak distribution is shown in the A2 and A3 regions outside A1. However, the resolution of the H distribution worsens as the distance from the center is increased. Initially, in the jpeg image, we thought that there would be no H value in the A4 region when viewed with naked eye. However, in the raw image, the H value was present although it was not a sharp H peak distribution, as shown in Figure 5e. R, G, and B values were mixed here. The background was placed at the bottom of the H distribution. Furthermore, for a longer exposure time, a larger saturated area is obtained, and this corresponds to the A1 region. The H distribution in this saturated region exhibits a tail instead of a normal distribution, as shown in Figure 5b. In other words, it is necessary to determine the selection area for the final analysis according to the H distribution.

### 4.2. Investigation Related to Intensity

(1)
*Exposure time and saturation*


In the case of jpeg images, it is impossible to compare the brightness or intensity of each image because it already has standardized RGB values. When a raw image is used, the maximum amount of charge is determined for each camera pixel. Therefore, when light of a certain intensity or more is incident, saturation occurs at a certain value. We checked for saturation according to exposure time. The Canon EOS 450D camera (manufacturer: Canon, Tokyo, Japan) used in this study exhibited the maximum charge value of 14,605 in each pixel, as shown in Figure 6. Therefore, a pixel value higher than this number can be considered saturated. As the exposure time was increased, the number of saturated pixels increased. Furthermore, near the laser light center, the color filter efficiency and the number of saturated pixels also increased.

(2)
*Exposure time and H value*


In the jpeg image case, the H distribution shifted and changed up to 30 completely depending on the exposure time or brightness because the H value changes during color conversion, white balance, or gamma correction. We checked that the peak H value showed a uniform distribution regardless of the exposure time in the case of raw images. However, a shorter exposure time resulted in a poor resolution of the H distribution. In the case of an excessively dark image, only the background could be seen. Figure 7 shows the images captured while changing the exposure time from 0.05 to 30 s.

(3)
*ISO number and H value*


ISO refers to the degree to which a signal is amplified during the digitization process [27]. Under the same conditions, the higher the ISO number, the brighter the picture. By adjusting ISO number, even in low light or dark places, we can take pictures with some brightness. However, a higher ISO number led to an equally higher noise. Variations in the H values were evaluated by changing the ISO values to 100, 200, 400, and 1600. For the pixels that had no saturated area and exhibited a certain brightness, the corresponding average H value for the entire pixel did not change significantly in the wavelength region from 400 to 650 nm.

## 5. Results

### 5.1. H-W Result Curve from LED

We used a Canon EOS D series camera with a CMOS image sensor in our study. Figure 8 shows the H-W mapping relationship using LED source for the wavelength from 400 to 650 nm. A monotonic decrease was evident, which is a typical characteristic of the CMOS CFA color filter in the H-W response curve, as mentioned in Section 2.1. Based on our LED inputs, the H value was changed with a smooth slope. Unlike the previous results that were mainly obtained in the G and R regions, as shown in Figure 1a, our measurement was expanded to the B region down to 400 nm. Due to the intrinsic limitations of the CMOS sensor, the camera was not sensitive below 400 nm [28]. Below this wavelength region, color images were not properly reconstructed. Therefore, special attention was required, or a camera dedicated to the ultraviolet (UV) region must be used in this wavelength region. Furthermore, over 650 nm, it was not possible to obtain the H-W relationship. The difference in conversion from H to W was excessively small such that it was difficult to distinguish the neighboring values. To measure this region, we need to use a camera with a sensor sensitive to the red or IR side. Figure 8 shows that overall the H-W relationship was highly non-linear, as we expected. The gray band represents the uncertainty due to the beam size we used and when the exposure time was doubled. In addition, much care was taken to reduce uncertainty caused by reflections on the screen. In order to reduce the error band, more collimated and higher-intensity sources should be used on the blue side. Below 400 nm, the error band is relatively large compared to that in the other ranges because of the poor camera response.

### 5.2. Investigation of the H-W Response Using Laser

Next, the determined H-W curve was cross-checked to assess its validity. The H values for several wavelengths were investigated using 375, 405, 440, and 473 nm laser modules. Among them, the 375 nm light source did not respond to the camera used in this study. At 405, 440, and 473 nm, the peak H values were approximately 240, 234, and 221, respectively, which remained unchanged irrespective of the exposure time. In addition, the output intensity of the 405 nm light was noticeably weak because of the intrinsic response of the camera. The three points matched the H-W curve very well within uncertainties, as shown in Figure 9.

## 6. Conclusions

We investigated the H-W relationship using a digital camera based on the CMOS sensor technology. To date, the H-W relationship in the red or green color bands has been relatively well-measured. However, no measurement results in the blue region or below 500 nm have been reported. Most of the signals generated in experimental particle physics or neutrino physics are read by a PMT, and the maximum QE of bi-alkali PMT lies in the blue region (400~450 nm). This is the reason why we are interested in the blue region of the wavelength and pay attention to the corresponding H-W relationship. There are two types of image sensors. One is CMOS CFA-based sensor technology, while the other is Foveon-based sensor technology. In our study, a camera employing CMOS sensors technology (Canon EOS D series) was used, and this was adequate for our purpose of using bi-alkali PMT-based experiment. Further investigations using Foveon-based cameras will be performed in the future.

If the RGB value of each pixel is known, then the HSV value can be obtained from this known RGB information. Subsequently, the H value can be converted to wavelength when the information contained in the HSV color space is converted to a visible light spectrum. An H-W curve was measured at wavelengths down to 400 nm using an LED; that is, the H-W measurement area was expanded down to the boundary between the visible and UV regions. However, our camera did not respond to the UV region. The response of the CFA of the CMOS sensor was obtained as an H-W mapping curve. A monotonously decreasing H-W curve is one of the main characteristics of the CMOS CFA-based image sensor. In addition, several factors affecting the H-W response were investigated.

In our future studies, this H-W relationship will be utilized to estimate as well as distinguish the color and wavelength of an image. Our method is anticipated to be used to obtain the emission wavelength information of a fluor dissolved in a liquid scintillator. By analyzing the images taken by digital camera or even mobile phone camera, it is possible to estimate the fluor emission spectrum in the visible wavelength region. This method can potentially replace the conventional expensive spectrophotometry techniques.

## Figures and Tables

**Figure 1 sensors-22-09497-f001:**
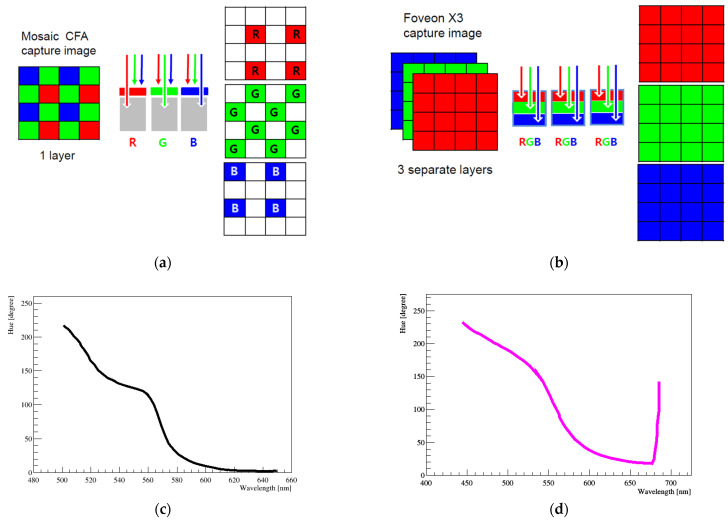
(**a**) Conventional mosaic captured using the CMOS sensor CFA technology. Color filters are applied to a single layer of pixel sensors and thus make a mosaic pattern. Only one wavelength of R, G, or B passes through a pixel, and one color is recorded. (**b**) Foveon X3 capture technology. The Foveon X3 image sensor has three separate layers of pixel sensors embedded in silicon. As a result, Foveon image sensors capture three colors at each pixel. (**c**) H-W response curve using CMOS CFA technology (for example, Canon 300D camera, manufacturer: Canon, Tokyo, Japan) adapted from [4]. RGB curves are input values to the camera. Plateau regions appeared at 530~560 nm and over 600 nm. (**d**) H-W curve based on the Foveon technology (for example, the SD 10 camera, manufacturer: Sigma Corporation, Bandai, Fukushima, Japan) adapted from [7,8]. The thickness of the line indicates uncertainties when the amount of light entering the camera is varied twice. The peak at the wavelength of around 675 nm is due to the poor camera response at the IR cut-off filter. All values beyond 675 nm are dumped to that point.

**Figure 2 sensors-22-09497-f002:**
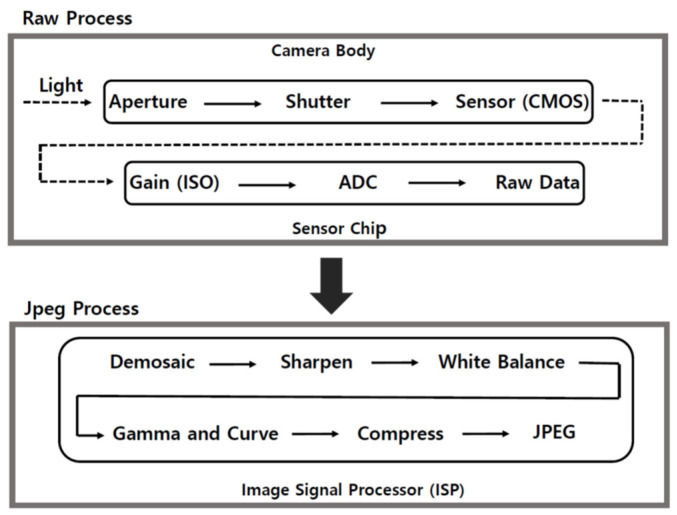
Schematic of the color image pipeline for raw and jpeg image. After the raw data are created, further processes for generating a jpeg digital image are continued. Each stage of the color image pipeline is fairly standard although varying in order or combined according to the manufacturers.

**Figure 3 sensors-22-09497-f003:**
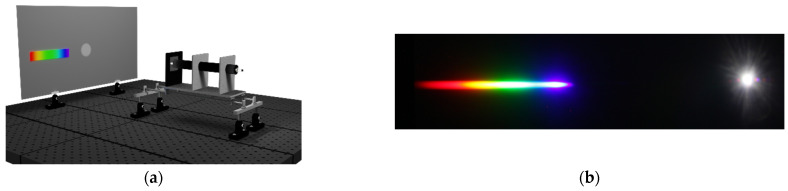
(**a**) Schematics of the experimental test bench set-up for the H-W curve measurement. LED light source, single slit (diameter 0.5 mm), collimator, diffraction grating (830 lines/mm), and screen were used. (**b**) One example picture showing the photographed diffraction patterns appearing on the screen. (**c**) Sketch of a distance (Y) measurement from the right central white image to the left each color band. From this value, we obtain the angle (θ) between the normal line of ray and the diffraction maxima.

**Figure 4 sensors-22-09497-f004:**
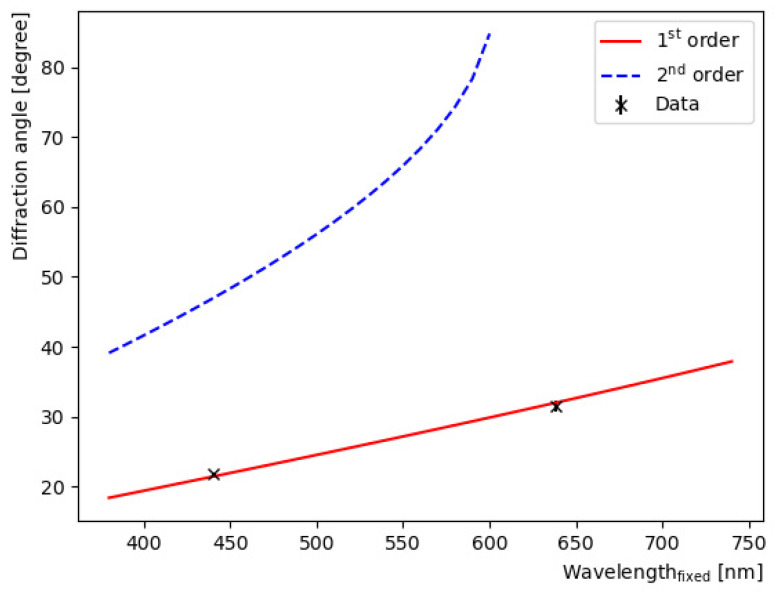
Comparison of the theoretical diffraction fringe position and laser point positions. For the theoretical diffraction curve, only the 1st order and 2nd order are drawn. The dotted line represents the 2nd-order diffraction pattern. In our experiment, only the 1st-order points were investigated.

**Figure 5 sensors-22-09497-f005:**
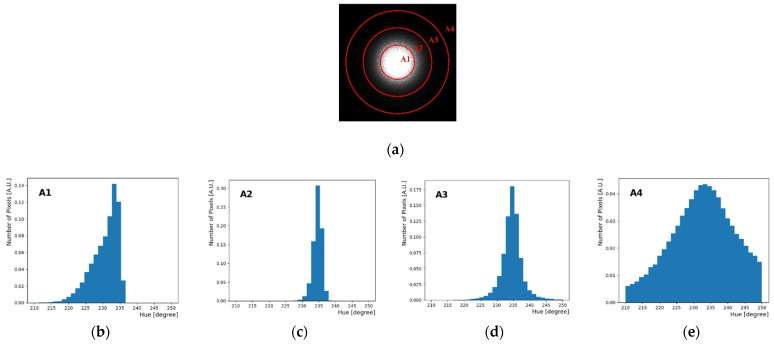
H distribution according to each area when using 440 nm laser source. (**a**) Raw image is divided into four circular areas (A1, A2, A3, A4). (**b**) H distribution of the innermost area 1. (**c**) Area 2. (**d**) Area 3. (**e**) The outermost area A4. The H peak positions are not significantly different (230.54, 234.55, 234.56, 235); however, the shape or resolution of the H distribution depends on the selected region.

**Figure 6 sensors-22-09497-f006:**
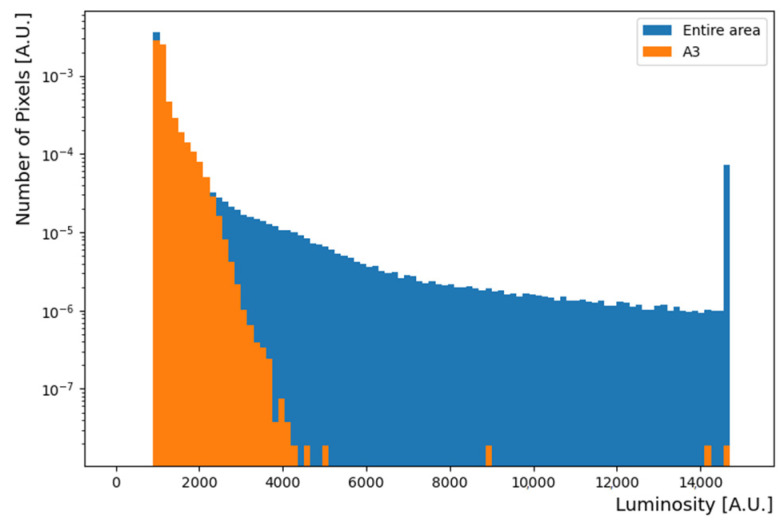
Charge distribution of all the pixels in the raw images. When its value was not normalized, the maximum charge value is 14,605 in each pixel. The minimum value for dark regions starts at 1000. The rightmost value is the case when a large amount of light enters. Blue color indicates the charge distribution of all the regions, and orange shows the charge distribution of the only the A4 region and sub sample of all the regions.

**Figure 7 sensors-22-09497-f007:**
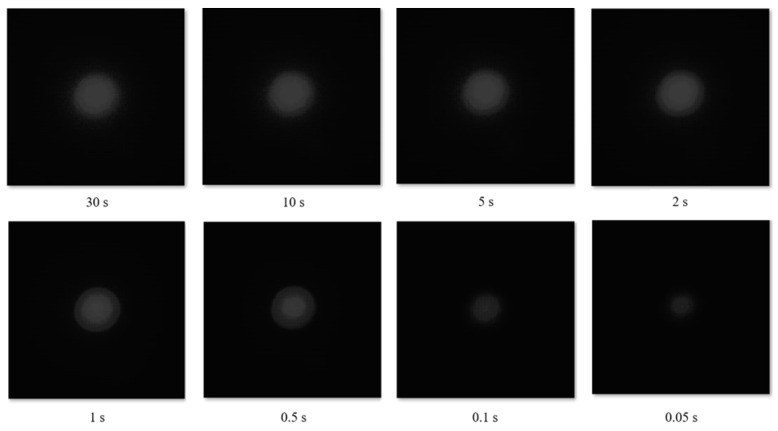
Raw pixel image change with exposure time at the wavelength of 440 nm. Raw images have a pixel value of 2^14^ scale. The exposure time is between 0.05 and 30 s.

**Figure 8 sensors-22-09497-f008:**
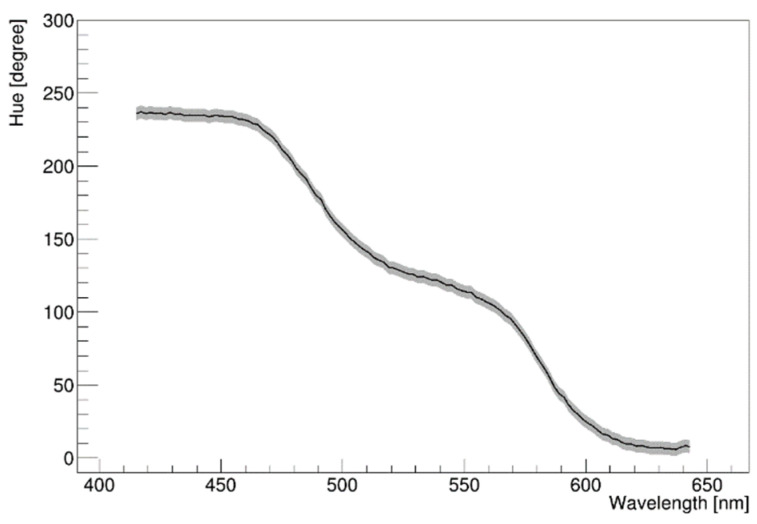
H-W curve obtained with Canon EOS D450 using the CFA technology for the wavelengths from 400 to 650 nm. The response of the CMOS camera is not sensitive outside this wavelength region. A monotonic decrease can be seen, and a non-linear H-W relationship is evident. The gray band represents an unavoidable uncertainty due to the beam size and doubled exposure time. In addition, the effect of reflection or dispersion on the screen also needed to be considered. The uncertainty in H was determined by standard deviation on the H distribution for a 2 nm scale wavelength.

**Figure 9 sensors-22-09497-f009:**
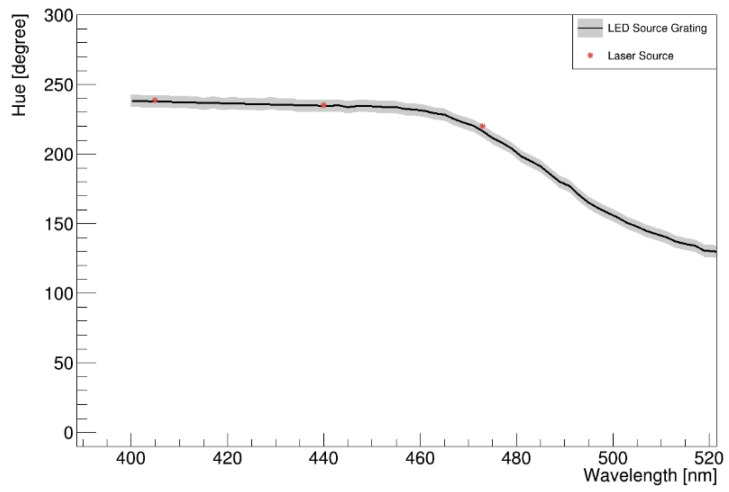
H-W response deduced using laser modules with wavelength 375, 405, 440, and 473 nm. Our camera did not respond to the 375 nm light source.

## Data Availability

Not applicable.

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
