# Peer review of "Investigation of the Hue–Wavelength Response of a CMOS RGB-Based Image Sensor"

_sensors, 2022, doi:10.3390/s22239497_

Round 1

Reviewer 1 Report

In this manuscript, a non-linear hue-wavelength (H-W) curve was investigated from 400 to 650 nm and declared that no study has reported on H-W relationship measurements, especially down to the 400 nm region. That's an interesting piece of work. However there are some minor comments below:

- even though no study reported the H-W relationship measurements  from 400 to 650 nm, the literature reviews should be included about H-W relationship measurements. 

-The authors mentioned that Numerous demosaicing algorithms have been proposed. Please provide the names of these demosaicing algorithms and give more details about the one shown in Figure 1 (a).  Is there any spesific name of the algorithm depicted in Figure 1 (a)

-Instead of 7.Summary, I would prefer "Conclusions" as 7. title.

-The summary part has to be extended in terms of demonstrating the clear impact. 

(no file added)

Author Response

Please take a look at an attached file. 

Thanks, 

Reviewer 2 Report

The topic of the paper is interesting as well as the results. Unfortunately, the paper is very short and the provided information is not deep enough to enable somebody to repeat this procedure of determining the H-W response of the CMOS sensor device.

I would so encourage the authors to invest more time in this paper and to explain the procedure in more detail. I would suggest including :
- the more detailed procedure on how to calculate H from raw RGB data;
- how the measurements of the distance from the central white image were made (somewhere the interval of 2 nm is mentioned);
- why the results (Figure 8) show an interval of H between 0 deg. and 240 deg. but in Figure 5 the H goes only from 210 deg. to 250 deg.
Besides that, it would be of great value if some pictures of a real setup were included.

Beside that:
- lines 132-133 the ISO is mentioned with its full name but it would be more understandable if "ISO number" or "ISO speed" would be used instead only "ISO",
- in line 208 Figure 1 is mentioned – is this the correct Figure? 

Author Response

(The authors gave the same response as above.)

Round 2

Reviewer 2 Report

Although the authors add some sentences and equations the paper could be made more interesting and more understandable if the authors would invest more time in it. The article could be greatly improved if chapter 3.1 was further expanded and explained how they define the position of the individual wavelength based on the image and how they get the H value, as well as how they determined the uncertainty they mention under Figure 8.

Author Response

Please take a look at an attached Q/A file. 

Thanks for your valuable comments one more time. 

Best regards, 

Joo
